# Professional Nurses’ Views and Experiences of Poor Health-Seeking Behavior Among Men in Limpopo Province, South Africa

**DOI:** 10.3390/healthcare12232320

**Published:** 2024-11-21

**Authors:** Lazarros Chavalala, Rachel Tsakani Lebese, Lufuno Makhado

**Affiliations:** Department of Public Health, Faculty of Health Sciences, University of Venda, Thohoyandou 0950, South Africa; rachel.lebese@univen.ac.za (R.T.L.); lufuno.makhado@univen.ac.za (L.M.)

**Keywords:** experiences, factors, health care, men, nurses, views

## Abstract

**Background/Objectives:** The health-seeking behavior of men is a public health concern and is associated with poor health outcomes and lower life expectancy among men. Masculinity norms are among the factors contributing to men’s poor health-seeking behavior. This study explored the views and experiences of purposively selected nurses on men’s health-seeking behavior. **Methods:** Through qualitative descriptive design, individual semi-structured interviews were conducted with 14 professional nurses, and collected data were audio recorded and transcribed verbatim. Tech’s eight steps were used to analyze data and guide the development of the main themes and sub-themes. Trustworthiness was ensured through credibility, confirmability, dependability, and transferability. Ethical approval was granted by the University of Venda Research ethics committee with reference number FHS/21/PH/26/1215. **Results:** Self-medication, a lack of trust in Western medicine, and the use of traditional healers were identified as reasons men underutilize health care services. Men were viewed as people who consult heath care services when illness is severe, feel uncomfortable with female nurses, and value respect from clinicians. **Conclusions:** Cultural norms still remain a barrier among men in this study setting, as men did not feel comfortable with their private parts being physically examined by female clinicians. There is a need to take note of cultural norms impacting health care service usage by men.

## 1. Introduction

Health-seeking behavior is any activity undertaken by individuals who perceive themselves to have a health problem or to be ill for the purpose of finding an appropriate remedy [1]. Globally, men continue to underutilize health services, and their poor health-seeking behavior has become a public health concern [2]. A study undertaken by Khajeh et al. [3] on health-seeking behavior has demonstrated that an individual’s behavior in engaging with health services is influenced by a variety of socioeconomic variables, such as age, sex, the type of illnesses, access to services, and perceived quality of the services. The delay in health care utilization has been associated with poor health outcomes and lower life expectancy among the male population [4]. With men being observed to have poor health outcomes, the attention of research has been drawn to understanding patterns of men’s health-seeking behavior and factors contributing to such behaviors. Studies have pointed out masculinity norms as one of the factors influencing the underutilization of health services among men [5]. Hawkins et al. [6] found that men’s beliefs about manhood and the desire to maintain control over their own health hinder their health-seeking behaviors. Some studies indicate that health care facilities have been reported as preventing men from accessing Human Immune Virus (HIV) services due to fears of confidentiality, as well as unsupportive and unfriendly responses toward them when seeking HIV-related services [7,8,9]. In South Africa, access to basic primary health care services is provided at no cost in public health facilities, yet men continue to underutilize the services. The South African National Integrated Men’s Health Strategy 2020–2025 was enacted to reduce the stigma associated with health-seeking behaviors. However, South African men continue to lag behind in health service utilization. This undermines the efforts made to freely provide access to basic primary health care services to all and support services provided to encourage men’s health-seeking behavior. Promundo and the United Nations Population Fund [10] state that access to free services remains a significant challenge to receiving quality health care in resource-limited settings and is a cause of underutilization of health care by men. Mthembu [11] indicates that most health interventions focus on women and children as a vulnerable group, while men are somehow excluded. Bibiano et al. [12] attest that men display more chronic health conditions and die more frequently than women from the main causes of death. Globally, 52% of all deaths recorded in 2012 from Non-Communicable Diseases (NCDs) were male, and males were more likely than females to die prematurely from NCDs in almost every country [13]. Taking note of Professional nurses’ (PNs) views and experiences when addressing factors contributing to poor health-seeking behavior among men could lead to better strategies to improve health-seeking behavior among the male population. For instance, a study conducted by Nkomo [14] in Mpumalanga, South Africa focused on the views of PNs regarding proposed national health insurance. In the current study setting, there are no known studies conducted and focused on the views and experiences of PNs on factors contributing to poor health-seeking behavior among men. Hence, this current study explored the views and experiences of PNs on factors contributing to men’s poor health-seeking behavior.

## 2. Materials and Methods

### 2.1. Study Design

The qualitative explorative design was employed to gather views and experiences of PNs on poor health-seeking behavior among men. The qualitative exploratory design was chosen for its ability to allow the researcher to dig deeper and gather detailed views and experiences.

### 2.2. Study Setting and Participants

In this study, views on factors contributing to poor health-seeking behavior and experiences of providing primary health care services to men were explored among PNs in Mopani, Vhembe, and Capricorn Districts of Limpopo province in South Africa. These districts were purposively selected for recording low health service utilization by men, in Limpopo province. There are 454 clinics, 26 community health centers, and 41 Hospitals, with some being specialized, tertiary, district, and regional hospitals in Limpopo Province [15]. There were about 13,701 registered PNs in Limpopo province in 2022 [16]. Researchers targeted professional nurses from Mopani, Vhembe, and Capricorn districts, and nurses were purposively selected. The study included professional nurses who had experience providing health services to male patients and worked within selected health facilities in the study setting. Professional nurses who did not have experience providing health services to male patients, worked in health facilities located outside the study setting, and those who refused to participate were excluded in the study. The sample size in the study was determined by data saturation, which was reached at 11, and an additional 3 individuals were further interviewed to ascertain if new information would come through (Table 1).

### 2.3. Data Collection and Analysis

The researchers conducted in-depth individual interviews using a semi-structured interview schedule containing the questions presented in Table 2 in May and June 2024. The questions were developed after a literature review, gaps identified, and research questions. The interview schedule was sent to experts in the field for review to ensure the credibility of the collected data. Interviews were conducted in English, because professional nurses were conversant with the English language. The interviews were conducted in health facilities and at residential places for nurses who were busy and could not receive a slot for an interview during working hours to avoid service disruption. The researchers made an appointment with professional nurses prior to interviews and visited facilities and agreed-upon homes as per set appointments. The interviews lasted between 20 min and 35 min on average. The interviews were conducted during convenient times for participants to avoid service provision disruption at health facilities. Researchers used eight Tech’s steps of data analysis to analyze raw data and guide the development of codes, sub-themes, and themes. Collected audio-recorded data were listened to and transcribed verbatim in Microsoft Word 365. Transcripts were then read in detail several times by the researchers, then codes were manually developed. Asked questions were checked against the developed codes, then themes and sub-themes were developed. Researchers coded independently, compared the codes, validated them, and reached a consensus on the developed codes.

### 2.4. Trustworthiness

Credibility was established through prolonged engagement with participants until data saturation was reached during interviews and independent data analysis by the three researchers. Researchers established dependability by being transparent on the selection of participants, data collection, and analysis of data documented in the study report. Confirmability was ensured through objective analysis of data and documentation of each step and decision taken during the study. Verbatim quotes were used to support the objective interpretation of data and show how true the researchers were in interpreting participants’ perspectives. For transferability, the researchers provided holistic descriptions of the study’s context, participants, sampling methods, and data collection sites to ensure the relevance of the findings to other settings.

### 2.5. Ethical Considerations

Permission to conduct the study in public health facilities was sought from the Limpopo Department of Health Research Ethics, District Departments of Health, in selected districts of Limpopo Province, and facility managers. Ethical approval was granted by the University of Venda Ethics Research Committee (Ethics Approval Number: FHS/21/PH/26/1215. Individual participants were given full information about the study including purpose, objectives, and procedures. Informed consent was obtained from individual participants before participating, and individuals were informed about their right to stop participating at any stage of participation in the study.

## 3. Results

Analyzed raw data led to the formation of two themes, nine sub-themes, and twenty-one categories on views and experiences of professional nurses on health-seeking behavior among men (Table 3). 

### 3.1. Main Theme 1: Views on Factors Contributing to Poor Health-Seeking Behavior Among Men

#### 3.1.1. Sub-Theme 1: Behavioral Factors

Self-medication and dealing with disease by themselves

The findings show that men self-medicate to address the health condition at hand and believe they can deal with the disease by themselves and be fine without consulting health facilities, especially public health facilities. The findings also show that men seek alternatives to deal with health conditions before they can opt to consult health facilities; if sought alternatives did not work, that is when they would consider a health facility.

Participants expressed views as follows:

“I think the one major thing that makes men not come to the facility is that they feel they can themselves deal with whatever personal issues or health issues that they have. So, a male patient would rather sort out the issue on his own” Clinician 1, Female, 34-year-old.

“Men can catch the flu now and do nothing, they think they can manage without medication” Clinician 2, male, 41-year-old.

“So most of them, they just go to traditional healers, and others they self-medicate at home” Clinician 6, female, 37-year-old.

Distrust of Western medicine and public health facilities

The findings show that clinicians view men as people who do not trust Western medicine and lack trust in public health facilities. African men believe Western medicine will not be effective and visiting public health facilities will not help them in finding solutions to their health problems. Clinicians indicated that men do not interact with health services from a young age, and that makes it difficult for them to develop trust in Western medicine and health services. The findings also show that men confirm diagnoses with private doctors.

Participants said:

“We’ve got people who believe in I’m just going to use traditional medication. I’m not going to take pills, because what are they? You know, it’s white people things” Clinician 3, female, 35-year-old.

“Most men will prefer to go to a private doctor than to come to a public facility, and I think it’s the thing to say they don’t trust that the public facility has got quality services. So, they tend to Prefer a doctor and like opinions from the doctor much better than the public facilities” Clinician 9, male, 43-year-old.

Fear of needles

The study findings show that men are afraid of being injected when they have visited health facilities as part of the medical intervention to address a condition that brought them to the facility. By staying away from visiting health facilities, they avoid being injected when faced with health conditions that require consultation at health facilities.

Participants said:

“They avoid injections because they don’t want their buttocks to be exposed or any other part of their body” Clinician 1, female, 34-year-old.

“I remember speaking to one patient, a male. He said that they are as afraid of needles as men. Hence that is why they don’t want to come to the clinic because they know that somewhere somehow, they are going to be injected “Clinician 5, female, 29-year-old.

Fear of knowing own health status

The study findings show that men fear knowing about their health status and avoid screening for diseases at health facilities. As a result, they screen for diseases through their partners and believe that if their partners test negative, they are also negative.

One participant said:

“So, they are just afraid of knowing their statuses. I remember someone saying that it’s better not to know than to know” Clinician 5, female, 29-year-old.

Consult traditional healers

The study findings show that men consult traditional healers when faced with health conditions. They believe that traditional herbs will make them recover faster than using western medicine. The results also show that men believe that they will have to take Western medicine for longer periods as compared to traditional herbs that can be taken for a short period and lead to getting better.

Participant said:

“Looking at an area being a rural area, there are many traditional healers. There are many herbalists, so most men, even some women, would rather seek help from a traditional healer or a herbalist than to go to the clinic” Clinician 9, female, 42-year-old.

Lack of Patience

The findings show that men are very impatient during attendance at health facilities. They cannot wait in long queues as they feel it is a waste of their time and expect to stay at the facility for a short while.

Participants said:

“First of all. Men are impatient. Men are impatient to sit in long queues” Clinician 8, female, 37-year-old.

“They are very impatient, so you don’t find them most of the time when you get to a facility waiting in queues. They even hate it when nurses go for tea, so they’re very impatient when it comes to that” Clinician 10, male, 43-year-old.

#### 3.1.2. Sub-Theme 2: Cultural Factors

The upbringing of men

The findings show that in black African cultures, from a young age, men are encouraged to be strong; as they grow and become adults, they hold a perspective of being strong beings. They take the masculine beliefs that surround their lives and apply the beliefs to their day-to-day lives. As a result, even when they are faced with health conditions that would require them to consult health facilities, they act strongly.

A participant said:

“They were raised told that you do things the traditional way. So, I think the most important thing is how the person is used to dealing with diseases” Clinician 6, female, 37-year-old.

Men are superior

The findings show that in black African cultures, men are seen as superior to women, and cannot accept help from women. The majority of men believe that they cannot be helped by female clinicians at health facilities as superiors. They feel embarrassed to engage in a helping session with female nurses, as their culture sets them out as superior. Therefore, they stay away from public health facilities as a way of avoiding meeting female nurses.

Participants expressed views as follows:

“In my experience, it would be about men feeling like they are superior and above, so they don’t want to come to a place where they need to meet a lot of women. They will be like, when we get there, we are being treated by females. So, they feel like women are inferior to them” Clinician 3, female, 35-year-old.

Upholding cultural values

The findings show that men feel uncomfortable having their private parts seen by female clinicians. In African black cultures, it is taboo to expose private parts to another person of the opposite gender who is not your partner. As a result, men avoid going to health facilities as a way of avoiding being helped by female clinicians who may require them to show their private parts for examination in cases where they have a condition that affects their private parts.

Participants said:

“When they feel like I’ve got an STI and I’m going to get there. I’m going to be seen by a female nurse. She’s going to ask me to undress, and I don’t want to undress in front of a person” Clinician 3, female, 35-year-old.

“It is difficult to tell female as a man what you’re experiencing, especially when it comes to either sexual matters or even a simple thing like a cough. They feel very intimidated, shy, or embarrassed talking to a female clinician. So, they are avoided by not coming to the clinic at all” Clinician 7, female, 40-year-old.

#### 3.1.3. Sub-Theme 3: Socio-Economic Factors

The stigma attached to visiting health facilities

The findings show that men do not feel comfortable being seen attending health facilities. They fear being judged and labeled by people who know them. They think that if someone sees them in the queue at a health facility, they will be labeled as a person with a specific health condition. As a way of avoiding being judged and labeled, men stay away from health facilities so that people do not know anything about their health issues.

Participants said:

“Immediately when people see somebody in the queue, it’s related to HIV. So, men don’t want that stigma. They don’t want people to know their business. They don’t want people to know that they have a chronic condition” Clinician 8, female, 37-year-old.

“There are some clinics there in a community whereby it is sort of utilized by neighbors. So, when a man comes to a clinic and the neighbors are there. They have got the stigma of saying, you know what, these people, they would think I have this type of a disease” Clinician 12, male, 47-year-old.

Peer influence

The findings show that men influence each other not to opt for public health care services when faced with conditions requiring medical attention, instead, they encourage each other to use other means such as consulting a traditional healer to address their health problems.

Participants said:

“So, with where they are staying, the influence would be around the people that they hang around With. The friends they keep, so a person would be sick and say I want to go there. Their friends would be like how come we don’t go to the clinic? You know? Why are you going there? We don’t get sick. And it would be some sort of a peer pressure that comes around with it” Clinician 3, female, 35-year-old.

“The first one I would say it’s pride and peer pressure because most men when they are sitting discussing things, are the ones who always say that they don’t believe in Western medicine. So most of them just go to traditional Healers.” Clinician 6, female, 37-year-old.

Employment

The findings show that men do not have time to attend health facilities due to work commitments. They spend most of their time busy at work and are sometimes employed in areas far away from health facilities. By the time they are knock off, health facilities are already closed, and they fear losing a job if they become absent to attend to health facilities.

Participant expressed views as follows:

“You get people who are working in firms and didn’t come to the clinic because of work. They couldn’t get time off.” Clinician 3, female, 35-year-old.

#### 3.1.4. Sub-Theme 4: Environmental Factors

Distance to health facilities

The findings show that in some areas, public health facilities are not easily accessible; service users, including men, have to travel long distances to access health facilities, and some may require transport.

One participant said:

“When you look at how they are living, most of them live in either informal dwellings or villages which do not have clinics and are very far from the clinics, so having to travel to the clinic tends to be a bit far for them to travel” Clinician 1, female, 34-year-old.

Easy access to traditional healers

The findings show that in some areas, traditional healers and traditional herbs are easily accessible compared to health facilities. As a result, men opt to consult traditional healers as they can easily access them. and are given herbs. compared to visiting health facilities that may require traveling.

A participant said:

“When you look at the street signs and the walls, there’s always a sign saying these are sold here, traditional healer at the next corner. Those are easily accessible compared to the clinic where you have to perhaps use a taxi or walk a bit of a distance” Clinician 11, female, 27-year-old.

### 3.2. Main Theme 2: Experiences of Professional Nurses in Providing Primary Health Care Services to Men During Treatment Sessions

#### 3.2.1. Sub-Theme 1: Health-Seeking Behavior Patterns

The findings show that men who visit health facilities to consult do so when they are in a severe state and focus on the disease at hand. They do not come for minor ailments and wait for the disease to be severe. The findings also show that men only visit when their partners are diagnosed with diseases and visit health facilities as the last option.

Participants expressed their experiences as follows:

“They come when they are already extremely sick and they have no other option because they’ve tried every other option” Clinician 5, female, 29-year-old.

“Men wouldn’t come to it clinic immediately. Men would want the problem to be severe” Clinician 2, male, 41-year-old.

#### 3.2.2. Sub-Theme 2: Dishonesty

The findings show that men lie about symptoms of illness that brought them to health facilities and do not open up immediately. They first observe the environment before becoming comfortable and opening up. They require a clinician to probe and be gentle with them to open up.

One participant said:

“Men tend to lie about their symptoms. So, for example, a man says when I pee, it burns, but as you probe, you find out that no, this man has scrotal swelling” Clinician 1, female, 34-year-old.

#### 3.2.3. Sub-Theme 3: Lack of Patience

The findings show that men appear very impatient during treatment sessions, in a hurry, and expect the session to end quickly so that they walk out of the consultation room. They avoid asking questions, minimally talk, do not concentrate, do not like to be questioned, and become angry when questioned. The findings also show that men dislike long treatment courses.

Participants said:

“They want something quick and fast, and they want to rush out of the consultation room” Clinician 11, female, 27-year-old.

#### 3.2.4. Sub-Theme 4: Valuing Respect

The findings show that men require respect and open up when they feel respected. They appreciate respect at the end of the session, and when they feel they were respected and are likely to come back for consultation.

“From the others that I’ve treated, you find that they’re very grateful after being serviced very well and treated with respect and dignity” Clinician 7, female, 40-year-old.

#### 3.2.5. Sub-Theme 5: Feel Uncomfortable with Private Part Physical Examination

The findings show that men refuse private part physical examinations by female nurses because they feel uncomfortable. A female clinician had to ask a male clinician to assist. The findings also show that men do not like physical examination due to uncomfortable feelings about being touched by clinicians.

Participants expressed their experiences as follows:

“They don’t want to expose their bodies because they don’t want you to see their private parts or what. It is the reason they lie about their symptoms” Clinician 1, female, 34-year-old.

“I’ve also had an experience with one patient whereby they came to the facility, did not want to open so that we can see, they were given a different type of treatment because well, we can’t see what’s going on” Clinician 3, female, 35-year-old.

## 4. Discussion

### Views of Professional Nurses on Factors Contributing to Poor Health-Seeking Behavior Among Men

The PNs viewed the conduct of men as among the factors that contribute to men’s poor health seeking behavior. Self-medication has been viewed as one of the reasons men avoid consulting at health facilities. This is supported by study findings of Mthembu [11] conducted in Durban, South Africa, on men, and it was discovered that men in his study reported diagnosing and treating themselves instead of visiting health facilities for consultations. A study conducted by Arumugam et al. [17] in Malaysia of males found that men self-medicated based on their cultural beliefs and understanding of the disease severity before engaging with health care services. These findings clearly show that when men successfully self-treat diseases, they end up not seeing the significance of engaging in health care services.

Fear of disease screening was also seen as among the reasons men avoid consulting public health facilities. This can be associated with a reason men screen for diseases through their partners as another reason identified by PNs. This behavior is an obstacle in achieving The Joint United Nations Programme on HIV/AIDS (UNAIDS) 95 95 95 strategy targets, where 95% of people living with HIV must know their HIV status; 95% of people who know their HIV-positive status are on treatment; and 95% of people on treatment have suppressed viral load [18]. As more men remain undiagnosed, they are likely to suffer from poor health outcomes, loss of life, and the spread of diseases. These findings are in line with the findings of Lim et al. [19] in Malaysia where male participants avoided HIV testing because of fear of positive results.

The use of traditional healing services has also been viewed as a reason men do not utilize public health services. This leads to further complications that are hard to manage at a latent stage of disease, especially for diseases that require medical interventions. This suggests a need to educate men about the importance of knowing diseases that can be addressed using traditional healing services. The use of traditional healing services may be linked to easy access to traditional healers, lack of trust in Western medicine, and public health facilities, which PNs viewed as other reasons men underutilize public health facilities. However, these findings differ from the study findings of Fish et al. [20], conducted in Australia among men, where it was found that men preferred clinicians over other forms of help-seeking and had a high level of trust in health professionals. This difference may be due to the cultural background the participants grew up in and how health issues are viewed.

Culture was also viewed as a factor contributing to poor health-seeking behavior among men. Masculinity beliefs contribute to men’s poor health-seeking behavior. It is evident from the findings that men who believe in their African cultural practices would not find it easy to consult at public health facilities as a female-dominated environment. Their cultural practices and values are not in favor of the Western methods used in the health sector. For instance, females being able to examine the private parts of male patients. This indicates the necessity of considering culture in the provision of health services. This is supported by the findings of Belli et al. [21], where a study conducted in Durban, South Africa among men found that men’s attempts to comply with cultural norms inhibit health-seeking behavior.

The stigma attached to using health facilities was also seen as a factor contributing to the underutilization of public health services. This means that there is still a myth held by community members about utilizing health services. This calls for a need to educate the public about services offered at public health facilities and change the way people think about those seen utilizing health services. These findings are supported by the findings of Cleary [22], who discovered that males in Ireland did not visit health care professions because of fear of being judged by peers for engaging with health care services.

Being employed has also been viewed as a driving factor contributing to poor health seeking behavior and underutilization of public health services. This means the health of employed people who are not able to receive time to visit health facilities is compromised. This suggests a need to collaborate with employers and find a way to allow employees to take care of their health.

Access to health services due to distance to health facilities has also been viewed as a contributor to poor health seeking behavior and low utilization of public health services among men. These findings are in line with the findings of Fish et al. [20] where men in Australia had difficulties accessing health care services due to the unavailability of medical professionals in rural areas. These findings clearly show that the unavailability of health care services in nearby places deter men from engaging with health care professionals. Difficulty in accessing health care services may be the reason men consider other treatment care alternatives to deal with the situation at hand.

Experiences of professional nurses in providing primary health care services to men during treatment sessions.

It is evident from participants’ experiences that the health-seeking behavior pattern of men is not improving, as men were reported to still consult late when the disease is already in a severe state. This explains why men are more likely to die from diseases compared to women. These findings are in line with the findings of Nzama [23] who found that males in Durban, South Africa delay seeking health care, hoping that the STI would go away, and only visit health facilities when the STI was severe.

PNs experiences also highlight the role culture plays in men’s health seeking behavior. This is reflected in experiences where men as patients refused physical examination by female clinicians because they did not feel comfortable with being physically examined on their private parts. These findings are supported by Lyu et al. [24], who conducted a study on male nurses’ experiences of providing intimate care to female patients in China. Their findings indicate that there was a low acceptance rate of services where female patients had to expose the perineum for catheterization or perineal irrigation. Another study conducted by Inoue et al. [25] on male nurses’ experiences with providing intimate care for women clients in Australia found that nurses found it challenging to provide intimate care to female patients. This clearly shows that feeling uncomfortable with a service provider where the service provider is of the opposite gender can make both men and women avoid visiting a health facility to seek help. These findings also highlight how cultural norms may affect service provision and its outcomes. A man who does not agree to be physically examined is likely to leave the facility untreated for the condition that brought them to the facility because the clinician does not know what the patient is dealing with. As a result, he is likely to lose confidence in the health system because he was not assisted with his problem. A study conducted by Layak et al. [26] on male student nurses’ experiences with care in Brunei Darussalam discovered that gender differences in providing care were not a concern because they were allocated in the male wards.

Lack of patience in men during treatment sessions was concerning for PNs. Men were reportedly in a hurry and expected the session to end quickly, avoided asking questions, and minimally talked. This study’s findings differ with findings of Faraji et al. [27]’s study conducted in Iran on nurses, which found that nurses complained of devoting more time to patients to provide more care and performing tasks they felt were unnecessary.

Giving men respect reportedly encouraged them to visit again and open up during treatment sessions. This means that when men are respected, they feel accepted and believe they will not be judged and find it easy to open up. Respecting patients can also encourage patients to utilize health care services. These findings are supported by Fernandez et al. [28] who found that female patients in South Floria felt respected when a physician did not put blame and judge them.

Based on this study’s findings, future research could investigate the association between easy access to traditional healers and underutilization of health services. Future researchers could also explore factors contributing to men’s lack of trust in the health system. The findings also implicate that clinicians’ respect for male patients during treatment sessions is crucial and encourages males to engage with health care services. This points out that there is a need to encourage clinicians to provide health services in respectful manner when dealing with male patients. There is a need for clinicians to strengthen clinician–patient relationships to encourage more males to interact with health care services. As the study pointed out, the behavior of men, such as the fear of knowing one’s own health status, trust issues involving western medicine, and dealing with diseases by themselves, contribute to men’s poor health seeking behavior; there is a need for behavior modification among men through education. The health sector should work hand in hand with community stakeholders such as traditional healers and community leaders in changing men’s behavior and actions when faced with health problems. To address stigma attached to using health services and masculinity beliefs and how men are raised as factors contributing to men’s poor health seeking behavior, there is need for community awareness on health service utilization, including services offered at health facilities. Masculinity beliefs and societal norms associated with how men are raised should be challenged through community education to give community members information on how norms and masculinity believes affect men’s health. As the study pointed out, regarding the use of traditional healers as an alternative to address health challenges by men, there is a need for collaboration between the health sector and traditional healers in providing health services. This would strengthen referral systems between traditional healers and the health sector in providing services. For policymakers, there is a need to incorporate black African culture when developing policies around health service provision to address cultural issues impacting the utilization of health services among males.

The researchers conducted the study as per the original design in the protocol and specific participants were selected to represent the targeted population. Researchers reported the study findings as they were without the influence of the funding party.

The study has potential limitations, there was a delay in obtaining permission to access health facilities from the Limpopo Provincial Department of Health Research Committee, thus, researchers had limited time to engage participants. Some participants were only available for interviews during working hours; hence, they avail themselves for a limited period, and interviews were held during the given period.

## 5. Conclusions

Men continue to lag behind in terms of seeking health care when faced with an illness and consult when the disease progresses to the latent stage and become less patient during their attendance at health facilities. Cultural norms remain a barrier among men in this study setting, as men did not feel comfortable with private part physical examinations by female clinicians. There is a need to take note of cultural norms impacting health care service usage by men. However, respecting male patients has been observed to encourage men to further engage in health care, and therefore highlights the importance of building strong working rapport with male patients.

## Figures and Tables

**Table 1 healthcare-12-02320-t001:** Participants demographic data.

Participant	Gender	Age	Years of Experience
P1	Female	34	6
P2	Male	41	10
P3	Female	35	9
P4	Male	31	7
P5	Female	29	7
P6	Female	37	12
P7	Female	40	15
P8	Female	37	13
P9	Female	42	16
P10	Male	43	14
P11	Female	27	5
P12	Male	47	16
P13	Female	36	12
P14	Female	43	10

**Table 2 healthcare-12-02320-t002:** Summary of main interview questions.

Summary of Key Questions
What do you think are the reasons men do not visit healthcare facilities to utilize healthcare services?
From your experience in providing primary health care services to patients, what have you observed among men during treatment sessions?

**Table 3 healthcare-12-02320-t003:** Main themes, sub-themes, and codes developed from results of individual interviews with Professional Nurses (PNs).

Main Themes	Sub-Themes	Codes
Views on factors contributing to poor health-seeking behavior among men	Behavioral factors	Self-medication and dealing with the disease by themselves, lack of trust in Western medicine and public health facilities, fear of needles, fear of knowing own health status, consulting traditional healers, impatience
	Cultural Factors	The upbringing of men, men are superior, upholding cultural values
	Socioeconomic factors	The stigma attached to visiting health facilities, peer influence, employment
	Environmental factors	Distance to health facilities, easy access to traditional healers
Professional Nurses’ Experiences in providing primary health care services to men during treatment sessions	Health-seeking Behavior Patterns	Consultation when the disease is in a severe state, not coming for minor ailments, and only visiting when their partners are diagnosed with diseases
	Dishonesty	Men lie about symptoms of illness
	Impatience	Men are always in a hurry and expect the session to end quickly
	Valuing Respect	Appreciate respect
	Feel uncomfortable with Private part physical examination	Men refuse private part physical examination

## Data Availability

Data are available on request.

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
