# Peer review of "Professional Nurses’ Views and Experiences of Poor Health-Seeking Behavior Among Men in Limpopo Province, South Africa"

_healthcare, 2024, doi:10.3390/healthcare12232320_

Round 1
Reviewer 1 Report
Comments and Suggestions for Authors
Dear Editor,
Thank you for the invitation to review this manuscript. This manuscript reports the findings of a qualitative study regarding nurses’ views and experience of poor health-seeking behavior among men in Limpopo province. The manuscript is well written and includes all essential information for a qualitative study. Some parts need clarification to enhance the clarity of this manuscript, as below:
- Title: Without intention to degrade the province, I suggest the authors mention the country where the study was conducted instead of the province, or add the country after the province. Please be informed that the readers of the journal are international readers, and some might not know where Limpopo province is.
- To describe the study setting and participants, the authors informed on page 2 last paragraph that “There were about 7743 registered PNs in Limpopo province in 2013," which was the data of eleven years ago. Please provide the recent data about the number of registered PNs in the setting.
- On page 2, in the last sentence of the second paragraph, it is written, “NCDs in almm,.ost every country." Do you mean “NCDs in almost every country”? Please revise the typing error.
- What are the inclusion and exclusion criteria of the participants? Please add the information.
- Who performed the interviews, and where did the interviews take place? Please add the information.
- Please move Table 1 (participants demographic data) from the method section to the result section.
- On page 5 last paragraph, when citing the excerpt from clinician 8, it is written only clinician 8. Please add gender and age after the information of clinician 8 as you write for other participants.
- Please add the strengths and limitations of the study.
- What are the implications of this study in the community and the recommendations for stakeholders and decision-makers? As found in this study, men did not want to visit healthcare facilities due to several factors, for example, fear of being judged and labeled by communities or getting influenced by other men. Those are important findings that need to be followed up in the real situation. Presenting the implications and recommendations based on the findings of this study can be one of the solutions and might help to reduce these issues in the community.
Author Response
Reviewer 1 line by line response.
Comment 1:Title: Without intention to degrade the province, I suggest the authors mention the country where the study was conducted instead of the province, or add the country after the province. Please be informed that the readers of the journal are international readers, and some might not know where Limpopo province is.
Response: Name of a country added
Comment 2: To describe the study setting and participants, the authors informed on page 2 last paragraph that “There were about 7743 registered PNs in Limpopo province in 2013," which was the data of eleven years ago. Please provide the recent data about the number of registered PNs in the setting.
Response: Recent data on number of registerd nurses in Limpopo province added, There were about 13701 registered PNs in Limpopo province in 2022
Comment 3: On page 2, in the last sentence of the second paragraph, it is written, “NCDs in almm,.ost every country." Do you mean “NCDs in almost every country”? Please revise the typing error.
Response: error corrected, NCDs in almost every country
Comment 4: What are the inclusion and exclusion criteria of the participants? Please add the information
Response: agree, Inclusion and exclusion criteria added. The study included professional nurses who had experience of providing health services to male patients and worked within selected health facilities in the study setting. Professional Nurses who did not have experience of providing health services to male patients, worked in health facilities located outside the study setting, and those refused to participate were excluded in the study.
Comment 5: Who performed the interviews, and where did the interviews take place? Please add the information
Response: Information added. The researchers conducted in-depth individual interviews using semi-structured interview schedule containing questions. The interviews were conducted in health facilities and at residential places for nurses who were busy and could get slot for an interview during working hours to avoid service disruption
Comment 6:Please move Table 1 (participants demographic data) from the method section to the result section.
Response: Done
Comment 7: On page 5 last paragraph, when citing the excerpt from clinician 8, it is written only clinician 8. Please add gender and age after the information of clinician 8 as you write for other participants.
Response: Done, Gender and age added. Clinician 8, female,37-year-old
Comment 8: Please add the strengths and limitations of the study.
Response: Strength and limitations added. The researchers conducted the study as per the original design in the protocol and the correct participants were selected to represent the targeted population. Researchers reported the study findings as they were without the influence of the funding party.
The study has potential limitations, there was a delay in obtaining permission to access health facilities from the Limpopo Provincial Department of Health Research Committee, thus, researchers had limited time to engage participants. Some participants were only available for interviews during working hours hence they avail themselves for a limited period and interviews had to stick to the given period.
Comment 9: What are the implications of this study in the community and the recommendations for stakeholders and decision-makers? As found in this study, men did not want to visit healthcare facilities due to several factors, for example, fear of being judged and labeled by communities or getting influenced by other men. Those are important findings that need to be followed up in the real situation. Presenting the implications and recommendations based on the findings of this study can be one of the solutions and might help to reduce these issues in the community.
Response: Done, implication and recommendations added: This point out there need to encourage clinicians to provide health services in respectful manner when dealing with male patients
As the study pointed out that the behaviour of men such as fear of knowing own health status, trust issues on western medicine and dealing with diseases by themselves contribute to men’s poor health seeking behaviour, there is a need for behaviour modification among men through education. The health sector should work hand in hand with community stakeholders such as traditional healers and community leaders in changing men’s behaviour and actions when faced with health problems. To address stigma attached to using health services and masculinity believes and how men are raised as factors contributing to men’s poor health seeking behaviour, there is need for community awareness on health services utilisation including services offered at health facilities. Masculinity believes and societal norms associated with how men are raised should be challenged through community education to give community members information on how norms and masculinity believes affect men’s health. As the study pointed out the use of traditional healers as an alternative to address health challenges by men, there is a need for collaboration between the health sector and traditional healers in providing health services. This would strengthen referral systems between traditional healers and the health sector in providing services.
Reviewer 2 Report
Comments and Suggestions for Authors
This is an interesting study which explored the views and experiences of PNs on factors contributing to men’s poor health-seeking behavior. The main question is whether this kind of a study may serve as a basis to "lead to better strategies to improve health-seeking behavior among the male population".
I would suggest adding to this study population of PN a group of male patients asking them the first question that was posed to the PN. This addition may have two purposes: 1. It will serve as a validation of the impression the PN have. 2. The direct answer of male patients will better fit to the purpose of this study.
Editing comments:
1. There are several incomplete sentences referring to quotations.
- For example: "A study undertaken by [3] on ….", " [10] state…", " [11] indicates that…", " study conducted by [14]..". This is not acceptable. I suggest completing these sentences by adding the name of the author in front of each reference number.
- This problem is intensified in the discussion. When the authors write (at the bottom of page 8): " A study conducted by [17] found that men self-medicated based on their cultural beliefs…". This way of writing repeat itself in most quotations along the discussion. We need to know at least where the quoted study was conducted, who was its study population. The additional information would support the discussion of similar results or opposite results in the quoted studies.
- At the bottom of page 5 there is a quotation of 'Clinician 8' without specifying age and sex of the responder.
- In the beginning of the discussion the authors mentioned a new abbreviation – HSB. Though they used the full term along the whole paper, as it appears there for the first time, it should follow with the full expression.
2. There are some typing mistakes:
- On page 2: "males were more likely than females to die prematurely from NCDs in almm,.ost every country [13]."
- At the bottom of page 4: So most of them, they just go to traditional healers, and others they self-medicate at home” Clinician 6, female, 37-year-old – there should be a Quotation mark at the beginning of the quotation.
- On page 6: "… men a seen as superior to women." – I suppose that the author meant to write 'men are seen".
3. The list of references should be written uniformly.
Author Response
Comment 1: There are several incomplete sentences referring to quotations.
- For example: "A study undertaken by [3] on ….", " [10] state…", " [11] indicates that…", " study conducted by [14]..". This is not acceptable. I suggest completing these sentences by adding the name of the author in front of each reference number.
- This problem is intensified in the discussion. When the authors write (at the bottom of page 8): " A study conducted by [17] found that men self-medicated based on their cultural beliefs…". This way of writing repeat itself in most quotations along the discussion. We need to know at least where the quoted study was conducted, who was its study population. The additional information would support the discussion of similar results or opposite results in the quoted studies.
- At the bottom of page 5 there is a quotation of 'Clinician 8' without specifying age and sex of the responder.
- In the beginning of the discussion the authors mentioned a new abbreviation – HSB. Though they used the full term along the whole paper, as it appears there for the first time, it should follow with the full expression.
Response: sentences have been completed as per the comment. Names of authors were added to familiarise the reader with who the authors are without referring to the reference list. Details on where quoted studies have been conducted and who the participants have been added. The age and gender of clinician 8 has been added and HSB abbreviation has been eliminated and the full phrase of health seeking behaviour has been used.
Comment 2: There are some typing mistakes:
- On page 2: "males were more likely than females to die prematurely from NCDs in almm,.ost every country [13]."
- At the bottom of page 4: So most of them, they just go to traditional healers, and others they self-medicate at home” Clinician 6, female, 37-year-old – there should be a Quotation mark at the beginning of the quotation.
- On page 6: "… men a seen as superior to women." – I suppose that the author meant to write 'men are seen".
Response: Typing mistakes have been corrected and quotation mark has been added.
Comment 3: The list of references should be written uniformly.
Response: The reference list has been revised using the American Chemical Society as per journal guidelines.
Round 2
Reviewer 2 Report
Comments and Suggestions for Authors
The authors accepted all my editing comments and did the necessary corrections. However, I haven't seen any reflection to my main comment, which was part of the 1st version review:
"The main question is whether this kind of a study may serve as a basis to "lead to better strategies to improve health-seeking behavior among the male population".
I would suggest adding to this study population of PN a group of male patients asking them the first question that was posed to the PN. This addition may have two purposes: 1. It will serve as a validation of the impression the PN have. 2. The direct answer of male patients will better fit to the purpose of this study."
I may assume that adding more interviewee to the study is a burden, but at least I expected to see something about this comment in the discussion.
Minor comment:
- On the 2nd line of page 3 there is written: "… and at residential places for nurses who were busy and could get slot". I assume that it should be: …and couldn't ….
Author Response
Comment: The authors accepted all my editing comments and did the necessary corrections. However, I haven't seen any reflection to my main comment, which was part of the 1st version review:
"The main question is whether this kind of a study may serve as a basis to "lead to better strategies to improve health-seeking behaviour among the male population".
I would suggest adding to this study population of PN a group of male patients asking them the first question that was posed to the PN. This addition may have two purposes: 1. It will serve as a validation of the impression the PN have. 2. The direct answer of male patients will better fit to the purpose of this study."
I may assume that adding more interviewee to the study is a burden, but at least I expected to see something about this comment in the discussion.
Response: true this was conducted with the purpose of coming up with strategies to improve health-seeking behaviour among the male population. Looking at the nature of its findings, it may serve as a basis to lead to better strategies to improve health-seeking behaviour among the male population. About adding male patients asking them what they view as factors contributing to health-seeking behaviour among the male population, would be great, However, since this study was a very large project, it also includes males who did not regularly or never used health care services, and the findings have validated the impression of professional nurses. Unfortunately, the male population findings have been consolidated on their own as another research paper. We are just not sure if that should be included in the discussion section and refer to the findings of the other study we conducted.
Comment: On the 2nd line of page 3 there is written: "… and at residential places for nurses who were busy and could get slot". I assume that it should be: …and couldn't …
Response: agree with comment: corrected